# Management of *E. coli* sister chromatid cohesion in response to genotoxic stress

Elise Vickridge[1,2,3], Charlene Planchenault[1], Charlotte Cockram[1], Isabel Garcia Junceda[1] & Olivier Espéli[1]

Aberrant DNA replication is a major source of the mutations and chromosomal rearrangements associated with pathological disorders. In bacteria, several different DNA lesions are repaired by homologous recombination, a process that involves sister chromatid pairing. Previous work in *Escherichia coli* has demonstrated that sister chromatid interactions (SCIs) mediated by topological links termed precatenanes, are controlled by topoisomerase IV. In the present work, we demonstrate that during the repair of mitomycin C-induced lesions, topological links are rapidly substituted by an SOS-induced sister chromatid cohesion process involving the RecN protein. The loss of SCIs and viability defects observed in the absence of RecN were compensated by alterations in topoisomerase IV, suggesting that the main role of RecN during DNA repair is to promote contacts between sister chromatids. RecN also modulates whole chromosome organization and RecA dynamics suggesting that SCIs significantly contribute to the repair of DNA double-strand breaks (DSBs).

[1] Center for Interdisciplinary Research in Biology (CIRB), Collège de France, UMR CNRS-7241, INSERM U1050, PSL Research University, 11 place Marcelin Berthelot, 75005 Paris, France. [2] Université Paris-Saclay, 91400 Orsay, France. [3] Ligue Nationale Contre le Cancer, 75013 Paris, France. Correspondence and requests for materials should be addressed to O.E. (email: olivier.espeli@college-de-france.fr).

All cells must accurately copy and maintain the integrity of their DNA to ensure faithful transmission of their genetic material to the next generation. DNA double-strand breaks (DSBs), single-stranded gaps (SSGs) and DNA adducts such as interstrand crosslinks (ICLs) are serious lesions that, if left unrepaired, are potentially lethal to the cell. DSBs, SSGs and DNA adduct repair involve homologous recombination (HR) pathways[1,2]. On the basis of current models, there are two major pathways for recombinational repair and homologous recombination in *Escherichia coli*[3]. The daughter strand gap repair pathway requires RecFOR, RecA and RuvABC gene products and the DSB-repair pathway requires RecBCD, RecA and RuvABC gene products[4]. The initial step of DNA damage repair by HR requires RecA loading on single-stranded DNA (ssDNA). It is achieved either by RecFOR on SS gaps, or DNA resection up to a *chi* site, by RecBCD on a DSB. RecA loading and strand invasion are essential for homologous pairing and regeneration of replication fork structures[5]. RecA protein bound to ssDNA triggers the autoproteolysis of LexA and the induction of many genes from the SOS regulon[6–8].

The DSB-repair pathway strongly relies on RecA-mediated pairing of the damaged DNA molecule with an undamaged copy serving as a template during the repair process, presumably the sister chromatid. In eukaryotes, during replication, cohesins keep the newly replicated sister chromatids together before segregation[9]. Cohesins have been shown to be important for DSB repair in G2 phase and post-replicative recruitment of cohesins has been observed at the site of the DSB[10–12]. However, the DSB-induced cohesion is not limited to broken chromosomes but occurs also on unbroken chromosomes, suggesting that cohesion provides genome-wide protection of chromosome integrity[13,14].

In bacteria, following replication, sister loci do not immediately segregate, and the duration of cohesion is controlled by the activity of topoisomerase IV (Topo IV)[15–17]. The role of Topo IV in the segregation of sister chromatids has led to a well-accepted, but yet undemonstrated model, involving precatenane links as the major post-replicative cohesion factor in *E. coli*. Using a site-specific recombination assay, we demonstrated that interactions and genetic exchanges between sister loci (sister chromatid interactions (SCIs)) are favored for a 10–20 min period following replication[16]. These SCIs rapidly decrease when replication is arrested but persist if Topo IV activity is impeded, suggesting that post-replicative topological links enhance genetic exchange between homologous regions.

Previously, the absence of identified cohesins and the progressive segregation of bacterial sister chromosomes following replication have suggested that homologous recombination in bacteria requires a genome-wide homology search. Recent studies have demonstrated that a site-specific DSB can be efficiently repaired using distant sister homology[18]. These processes correlate with the formation of a RecA bundle and the merging of sister foci. In another study, DSB formation by a replication fork encountering a frozen topoisomerase provokes the rapid association of large regions of the previously segregated sister chromatids[19].

The SOS-inducible *recN* gene, which encodes an SMC (structural maintenance of chromosomes)-like protein, was identified over 20 years ago[20,21]. Expression of the *recN* gene is regulated by the LexA repressor, and following derepression, the RecN protein is one of the most abundantly expressed proteins in response to DNA damage[7,22]. RecN is also involved in the RecBCD-dependent DSBR pathway[21,23,24]. *recN* mutants are sensitive to ionizing radiation, I-SceI cleavage and mitomycin C (MMC)[24,25] but do not exhibit extensive DNA degradation following DSB[25]. *In vitro* assays have been developed with RecN from *Deinococcus radiodurans*. *D. radiodurans* RecN enhances ligation of linear DNA fragments suggesting DNA end bridging

or cohesin-like activities[26,27]. In addition, *Bacillus subtilis* RecN, which is among the first actors to the site of a DSB, promotes the ordered recruitment of repair proteins to the site of a lesion[28,29]. Interestingly, a different activity has been observed for RecN in *Caulobacter cresentus* and *E. coli*. It has been reported that RecN in this system is implicated in nucleoid dynamics following DSB repair[30,31].

Considering the sister chromatid cohesion and segregation mechanism in *E. coli* and the intriguing but unclear role of the SOS protein, RecN, we sought to investigate the importance of DNA precatenane-mediated sister chromatid cohesion in DNA repair. In this study, we used genotoxic agents to evaluate the role of topological links between sister chromatids in the repair of DNA damage. Our results demonstrate that SCIs are preserved upon treatment with MMC. Upon MMC treatment, SCIs become dependent on the induction of the *recN* gene product by the SOS response. Interestingly, a *recN* deletion can be fully rescued by a thermosensitive mutation in Topoisomerase IV, suggesting that the main function of RecN during DNA repair is to maintain SCIs, as precatenanes do under normal conditions. The loading of RecN onto sister chromatids is dependent on the presence of DSBs processed by RecA. Therefore, RecN can be considered as a DSB-specific cohesion factor. Because the presence of RecN accelerates growth resumption following genotoxic stress and affects the shape and dynamics of RecA repair structures, we propose that RecN-mediated preservation of SCIs is a key element in the repair of DSBs.

## Results

**SCIs are preserved in MMC-treated cells**. SCIs are essential for genomic stability. During the bacterial cell cycle, SCIs are determined by the balance between the rates of chromosomal replication and segregation[16] (Fig. 1a). We have previously developed a system that detects and accurately measures sister chromatid cohesion *in vivo*[16]. This *LacloxP* assay is based on the Cre-*loxP* site-specific recombination systems of bacteriophage P1. We engineered a cassette containing two adjacent *loxP* sites that can only recombine when Cre encounters the homologous region on the sister chromatid. The frequency of Cre recombination events is therefore dependent on the proximity between sister loci[16]. *LoxP* recombination was used in this study to monitor the organization and dynamics of sister chromatids following DNA damage induced by MMC. MMC is a potent antibiotic that inhibits DNA synthesis by reacting with guanines of complementary DNA at CpG sequences creating interstrand-crosslinks[32]. MMC treatment (i) promotes the formation of replicative lesions: SSGs[33] and DSBs[34]), (ii) induces the SOS response[35], (iii) blocks the completion of *oriC*-dependent chromosome replication (Supplementary Fig. 1A) and (iv) rapidly halts DNA replication (Fig. 1b,c). We placed the *loxP* cassette at the ori-3 locus (positioned 450 kb from *oriC*) and used this assay to measure SCIs in the presence and absence of MMC. To stimulate recombination, 20 min pulses of Cre induction were performed before MMC addition, immediately after MMC addition (0–20 min), 20 min after MMC addition (20–40 min) or 40 min after MMC addition (40–60 min). The recombination frequency slightly increased after MMC addition (Fig. 1d). This observation is in sharp contrast with the abrupt drop of the recombination frequency observed in a *dnaC* allele (*dnaCts*) when initiation of replication is blocked at a non-permissive temperature (40 °C; Fig. 1d). Interestingly, in the presence of MMC, SCIs also persist in the *dnaCts* strain at a non-permissive temperature (Fig. 1d). These observations suggest that MMC impedes sister chromatid segregation and renders SCIs independent of replication.

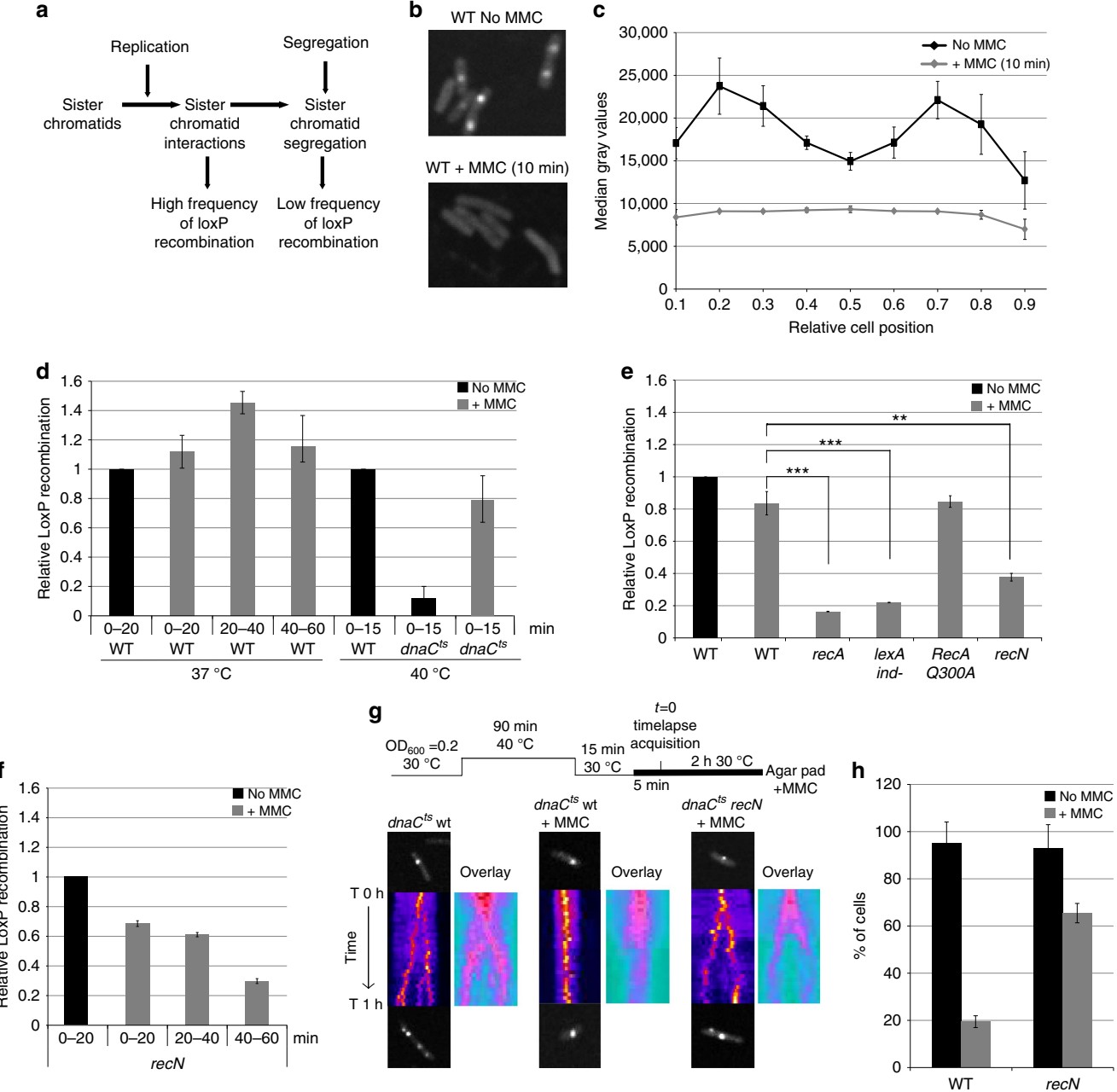

**Figure 1 | SCIs are preserved in the presence of MMC via a RecN-dependent pathway. (a)** The number of SCIs is under the control of replication that produces cohesive sister chromatids and segregation that separates them. **(b)** MMC causes rapid replication arrest. An MG1655 wild-type strain was used to monitor EdU incorporation in the presence and absence of MMC (10 μg ml$^{-1}$) for 10 min. **(c)** Quantification of a 10 min EdU incorporation pulse in WT cells treated or not with MMC. **(d)** SCIs were estimated at different time points after replication arrest in WT and $dnaC^{ts}$ cells treated or not with MMC. SCIs were measured in WT and $dnaC^{ts}$ strains by adding arabinose during 20 min pulses after MMC addition. In the $dnaC^{ts}$ strain initiation of replication was arrested at 40 min. **(e)** Upon MMC treatment, SCIs are dependent on RecA, LexA and RecN. SCIs were estimated using the frequency of loxP/Cre recombination at the ori-3 locus. The results at 40 min are presented; the results at 10, 20 and 30 min are presented in Supplementary Fig. 1. The results are expressed as the relative loxP recombination normalized to the untreated WT strain. Statistical comparisons of histological data were performed using Student's t-test. P values are considered to be significant for α = 0.05. *P < 0.05, **P < 0.01, ***P < 0.001. **(f)** SCIs are not maintained in a recN mutant. MMC was added and SCIs were measured as described in **d**. **(g)** MMC prevents sister chromatid segregation in synchronized cells. $dnaC^{ts}$ and $dnaC^{ts}$ recN cells tagged with a $parS^{pMT1}$/ParB-GFP system at ori-3 were synchronized for 90 min at 40 °C, and replication was re-initiated at 30 °C for 10 min. Cells were then placed on an agar pad, with or without MMC, and a time course analysis was performed. The images represent kymographs of a single-cell (fire colour map) and an overlay of about 20 cells (ice colour map). **(h)** Quantification of segregation events in $dnaC^{ts}$ WT and recN strains (100 cells were analysed for each strain). Cells were treated as described in **g**. Error bars are s.d. of four experiments.

## SCIs are dependent on RecA and RecN in the presence of MMC.

We sought to determine the role of HR and the SOS response in SCIs. Indeed, HR and more particularly the repair intermediates such as holliday junctions could promote SCIs. We observed that the degree of SCIs in the presence of MMC was strongly reduced in the absence of RecA (Fig. 1e, Supplementary Fig. 1B). This observation could suggest that RecA is required for preserving these interactions or that DNA degradation happening in the recA

mutant could more dramatically affect sister chromatids that are interacting (that is, the closest to the replication fork) than segregated sister chromatids. We used EdU incorporation to monitor DNA degradation. A brief incorporation of EdU before addition of MMC allowed us to measure degradation of the newly replicated DNA regions. We estimate that EdU labelling extended over 500 kb during the pulse, a distance that corresponds to the region involved in SCIs. In the WT strain, EdU foci were present in every cell for more than 3 h after MMC addition (Supplementary Fig. 1C,D) and the average fluorescence intensity of EdU labelling slowly decreased over the 3 h time course (Supplementary Fig. 1E). In some cells fluorescence intensity of EdU labelling increased at later time points (2 and 3 h). This increasing intensity is correlated with a reduction in the number of foci and a compaction of the nucleoids (Supplementary Fig. 1C). In the absence of RecA, the reduction of EdU intensity and the number of cells presenting EdU foci is only manifest after 1 h of treatment (Supplementary Fig. 1D,E). However, since DNA degradation was not significant during the first hour after MMC addition it was not correlated with the SCIs reduction observed in the first 40 m following MMC. Interestingly, in a mutant that is unable to induce the SOS response (lexA ind-)[36], we also observed a reduction of SCIs and loss of viability (Fig. 1e, Supplementary Fig. 1B,F). In order to test the relationship between the defects in SCIs and the ability of a strain to perform HR, we measured SCIs in the recA[Q300A] mutant. This mutant is strongly deficient in HR (<10% of the activity of a WT strain) but exhibits WT levels of LexA cleavage activity and SOS upregulation[37]. We observed that, despite a dramatic drop in the viability upon MMC treatment (Supplementary Fig. 1G), SCIs were maintained at a high level in the recA[Q300A] mutant compared to the recA mutant (Fig. 1e). Altogether, these observations indicate that completion of DNA repair and efficient HR are not required for the preservation of SCIs. Therefore we considered that a member of the SOS regulon may be responsible for SCI preservation upon MMC treatment.

To identify the SOS proteins responsible for maintaining SCIs, we analysed 15 mutants of SOS-inducible genes which functions has not been clearly established, for their ability to preserve SCIs after DNA damage. We observed that of the 15 mutants tested, only a recN mutation significantly reduced SCIs in the presence of MMC (Fig. 1e, Supplementary Fig. 1H). In good agreement with a low basal level of expression (Supplementary Fig. 2A,B), a recN deletion had no effect on SCIs under normal growth conditions (Supplementary Fig. 1B). We measured the persistence of SCIs in the absence of RecN using pulses of Cre induction, as described in Fig. 1d. LoxP recombination progressively decreased during the 60 min following MMC addition (Fig. 1f), suggesting that SCIs disappear in the absence of RecN. In good agreement with the fact that RecN is only expressed after RecA loading on ssDNA, the recN mutant did not present any DNA degradation phenotype upon I-SceI cleavage[25] or MMC treatment (Supplementary Fig. 1C–E). In the recN mutant, EdU foci degradation was comparable to WT cells even at the latest time points. Therefore the decrease of loxP recombination observed in the recN strain is not the consequence of DNA degradation. We favour the hypotheses that RecN is either involved in preserving SCIs or in the formation of de novo SCIs in the presence of MMC. Interestingly, the level of SCIs in the absence of RecN was significantly higher than in the recA or lexA ind- mutants, suggesting that other processes are altered in these mutants and perhaps an additional, yet-unidentified, SOS-induced protein participates in maintaining SCIs.

**RecN impedes segregation of damaged sister chromatids**. Our observations suggest that in the presence of MMC, RecN is required to maintain SCIs. To examine whether RecN is directly capable of preventing segregation of sister chromatids, we performed live cell imaging of sister chromatid dynamics. We utilized strains containing a dnaCts allele to synchronize the replication cycle and treated the cells with or without MMC 5 min after the estimated replication of a parS/ParB-GFP tag at the ori-3 locus. Replication synchrony and timing were measured by marker frequency analysis[38]. In the absence of MMC, both WT cells and recN mutant cells presented a reproducible segregation pattern in the 20 min following replication (Fig. 1g,h). In the presence of MMC, segregation was only observed in 20% of WT cells, the remaining cells contained either a single focus for the entire time course or brief alternating cycles of duplication and merging back of sister foci. In contrast, the majority (65%) of recN mutant cells presented a separation of the initial focus into two foci. These observations demonstrated that RecN is able to limit segregation of sister chromatids in the presence of MMC and therefore functions as a DNA damaged-induced cohesion factor in E. coli.

**The lack of RecN is compensated by extensive precatenation**. It has previously been demonstrated that sister chromatids stay cohesive behind the replication fork, forming structures known as precatenanes[16]. The type II Topoisomerase IV is responsible for the decatenation of these structures and ensures the correct segregation of both sister chromatids. We used a Topo IV thermosensitive mutant (parEts), in both WT and recN backgrounds, to test whether preventing the removal of precatenanes can maintain SCIs and restore viability in the presence of different DNA damaging agents. In untreated conditions, we observed an increase in post-replicative SCIs when the parEts mutant was shifted to a non-permissive temperature[16]. Following treatment with MMC, the frequency of SCIs decreased in the parEts cells but remained at a greater level than that observed in WT cells. Interestingly, the level of SCIs following MMC treatment was unaffected by either a recA or recN deletion in the parEts mutant (Fig. 2a, Supplementary Fig. 3A–F). This suggests that, in this context, maintaining precatenanes behind the replication fork can compensate for the absence of RecN, and that more generally; in spite of a DSB most precatenane links do not immediately disappear.

**Preservation of SCIs is linked to MMC-treated cell survival**. To assess whether the preservation of SCIs observed in the parEts recN mutant facilitates efficient DNA repair and cell survival upon MMC treatment, we performed CFU measurements in WT, recN, recA, parEts, parEts recN and parEts recA strains in the presence of MMC (Fig. 2b). Following a brief period at a non-permissive temperature, the recA and recN cells were strongly sensitive to MMC but the parEts cells were only slightly sensitive when compared to WT cells. Interestingly, the double recN parEts mutant presented a similar viability to the parEts mutant, suggesting that the decrease in viability in the recN mutant can be compensated by an increase in topological linking between sister chromatids during replication. On the other hand, inhibiting TopoIV in a recA mutant did not rescue viability, suggesting a specific role for RecN in the maintenance of SCIs (Fig. 2b and Supplementary Fig. 3G). We also measured the CFU in a parEts lexA ind- strain where the SOS response is down but RecA is present in basal levels (Supplementary Fig. 3H). The inhibition of TopoIV in a lexA ind- strain did not rescue the loss of viability of the lexA ind- strain. These observations strengthen the hypothesis that topological links, when they are artificially maintained, specifically compensate for a lack of RecN and therefore suggests that RecN is playing a structural role by maintaining sister chromatids close together.

**RecN requires DSBs and RecA to load on DNA**. To test whether the overproduction of RecN in a wild-type strain could affect SCIs, we constructed a vector containing RecN under control of a leaky promoter (pZARecN). Overexpressing RecN in the *recN* mutant restored viability and preserved SCIs following MMC

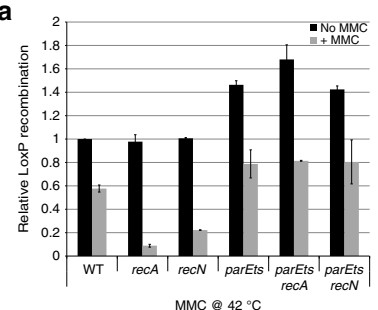

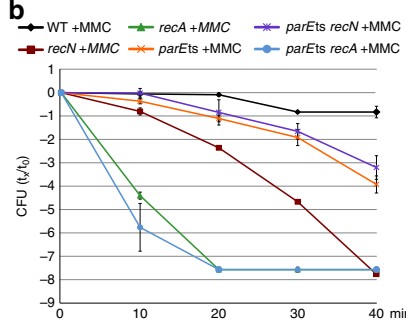

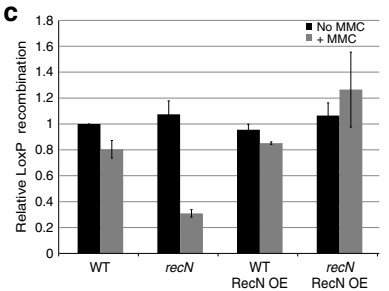

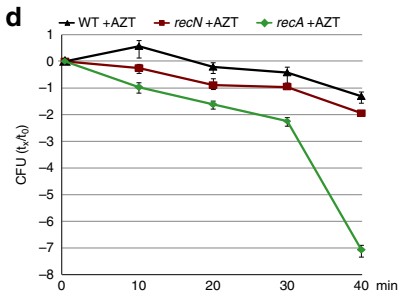

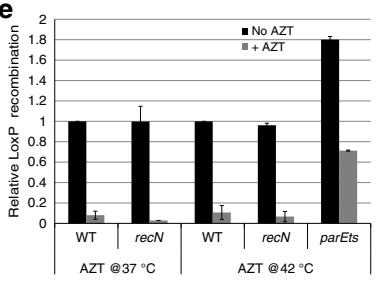

treatment (Fig. 2c, Supplementary Fig. 4A–D). However, RecN overexpression did not modify SCIs in untreated WT cells, in contrast to the Topo IV *parEts* mutation. These observations suggest that RecN is inactive in the absence of DNA lesions. To determine whether RecA itself or another SOS-inducible protein is responsible for RecN activation, we constructed an SOS constitutive strain (*sfiA lexA51*) in which *recA* is deleted but the SOS induction is maintained[39]. The *sfiA lexA51 recA* strain presented strong sensitivity to MMC (Supplementary Fig. 4E). In the *sfiA lexA51 recA*, we observed a low frequency of recombination, suggesting that DNA damage-induced SCIs are directly dependent on RecA (Supplementary Fig. 4F). We performed co-immunoprecipitation experiments in WT strains expressing RecN-Flag. In the presence of MMC, RecN was robustly co-immunoprecipated with an anti RecA antibody. Co-immunoprecipitation of RecA with an anti Flag antibody was less specific. Nevertheless in the presence of MMC and RecN-Flag induction the amount of co-immunoprecipitated RecA significantly increased (Supplementary Fig. 5). These observations demonstrate an interaction between RecA and RecN and that perhaps RecA serves as a loader for RecN. Interestingly, immunoprecipitation experiments revealed that a small amount of RecN was present even in the absence of MMC (Supplementary Fig. 5). We cannot distinguish if it corresponds to a basal level of RecN in all cells or to a fraction of cells inducing RecN through SOS in response to spontaneous damages. Considering the first hypothesis, this would suggest that RecN could intervene very early following DNA damage. To test whether RecA bound to SS DNA was sufficient to observe RecN-mediated SCIs, we used Azidothymidine (AZT) which is a DNA chain terminator. RecA is loaded on SS gaps in the presence of AZT; however, we observed no requirement of RecN on viability or SCIs in these conditions (Fig. 2d,e). SCIs disappeared rapidly in the presence of AZT in WT strain but were kept at high level when Topo IV was inhibited. This suggests that in these conditions precatenanes are rapidly removed and that RecN does not participate to SCI near SS gaps.

**RecN promotes the regression of segregated chromosomes**. Our data have demonstrated that RecN can prevent the segregation of newly replicated sister chromatids. However, it has recently been reported in *C. crescentus*, that RecN participates in DNA dynamics following the regression of segregated loci in response to an I-SceI-induced DSB[30]. We used time-lapse microscopy to evaluate RecN's putative role in sister chromatid dynamics following an MMC-induced DSB. We used a parB[pmT1]-YFP fusion that binds to a *parS* site inserted at the ori-3 locus

**Figure 2 | RecN-mediated SCIs are specifically established in response to DSBs.** (**a**) Measurements of SCIs following Topo IV alteration in the presence of MMC. *LoxP* assays were performed at 10, 20, 30 and 40 min after the addition of MMC. The results at 30 min are presented; the results at 10, 20 and 40 min are presented in Supplementary Fig. 3. The results are expressed as the relative *loxP* recombination, with MMC normalized to untreated WT. Cells were incubated for 25 min at 42 °C before the addition of 0.1% arabinose and MMC. (**b**) Influence of Topo IV alteration on WT, *recN* and *recA* mutant viability in the presence of MMC. The cell viability assay was performed at non-permissive temperature of 42 °C. (**c**) Influence of RecN overexpression on SCIs. The plasmid pZA31 carries *recN* under the control of a leaky promoter. The results are expressed as the relative *loxP* recombination of the MMC-treated sample normalized to the untreated WT. (**d**) Viability of WT, *recN* and *recA* mutants in the presence of SS gaps formed by AZT. (**e**) Measurement of SCIs in the presence of SS gaps. *LoxP* assays were performed as described in A. Error bars are s.d. of four experiments.

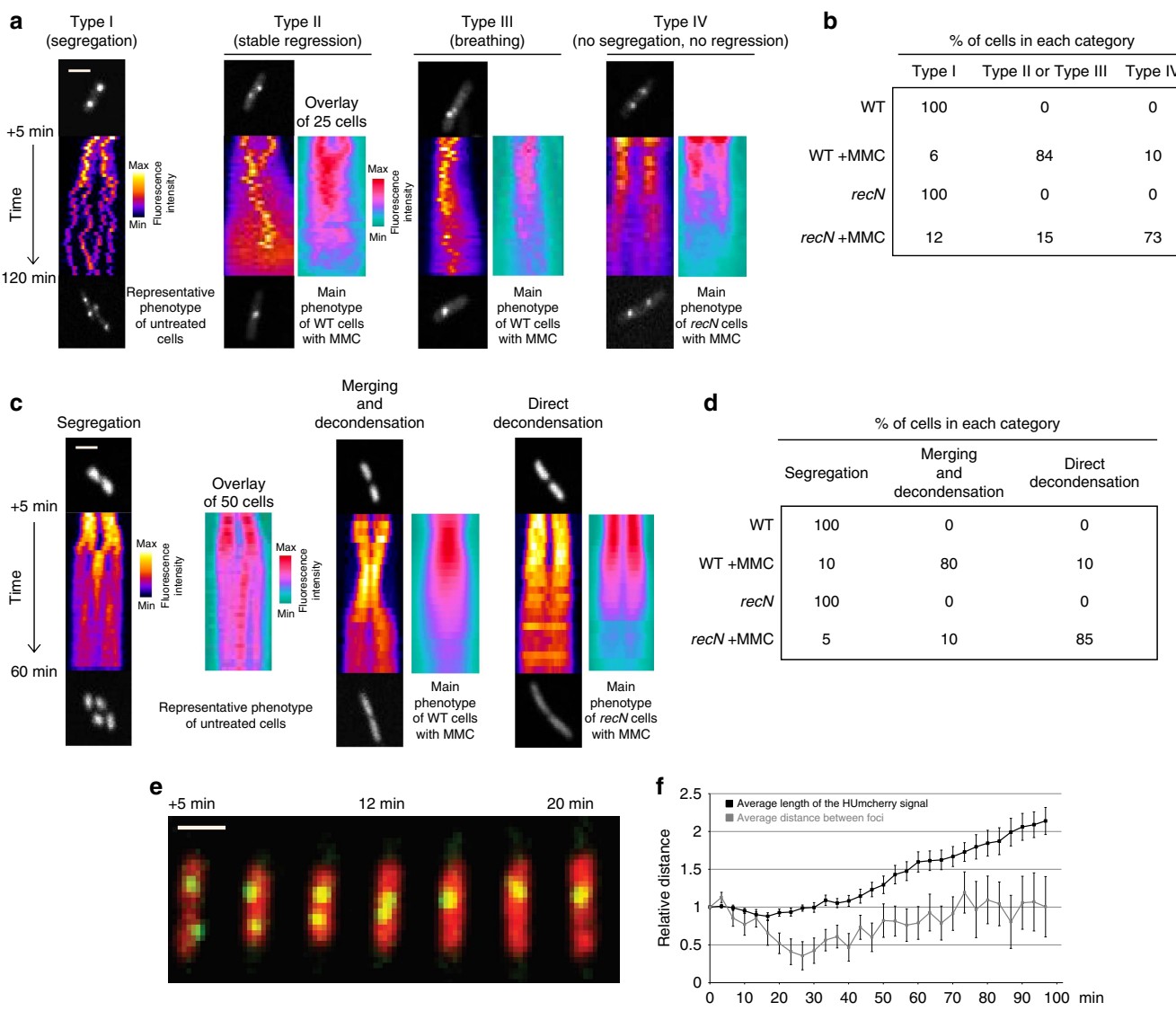

**Figure 3 | RecN participates in sister locus re-pairing and nucleoid rearrangement in response to DSBs.** (**a**) Representative kymographs of sister focus dynamics in the absence or presence of MMC. Kymographs were constructed along the long axis of the cell. Time-lapse imaging starts 5 min after the initial contact with MMC. Images were acquired every 3 min for 2 h. The fire lookup kymographs represent a single cell. The Ice lookup kymographs are an overlay of kymographs. (**b**) Frequency of the different types of sister focus dynamics was measured in the presence of MMC. Results are expressed as a percentage. About 100 cells were observed. (**c**) Representative kymographs of nucleoid (HU-mCherry) dynamics in the absence or presence of MMC. (**d**) The frequency of the different types of nucleoid dynamics was measured in the presence of MMC for the WT strain and the *recN* mutant. Results are expressed as a percentage. About 100 cells were observed. (**e**) Dynamics of sister foci and nucleoids in the presence of MMC. A time course was performed in a strain tagged with a *parS^{pMT1}*/ParB tag at ori-3 and labelled with HU-mCherry. (**f**) The distance between nucleoid edges and the distance between sister foci were recorded at each time point of the experiment presented on **e**. Distances were normalized to 1 for the time +5 min after MMC addition and are an average of 50 cells. Error bars are standard deviations of 50 cells. Scale bars are 1 µm.

(450 kb from *oriC*). In the absence of MMC, at time 0, more than 80% of WT cells contained two foci and we observed the segregation of the two foci into four foci immediately before cell division, each daughter cell containing two new foci (Fig. 3a, Type I and Fig. 3b). The presence of MMC significantly altered focus segregation. In most cases, bacteria with two foci at time point 0 failed to produce four foci, and eventually, the two segregated foci regressed back into one central focus that either persisted in this state for an extended period of time (Type II, 45% of the population) or regressed transiently into one focus (Type III, 39%; Fig. 3a,b and Supplementary Movie 1). This phenotype was strongly dependent on RecN (Fig. 3a,b and Supplementary Movie 2). We observed the beginning of regression between 12 and 47 min after MMC treatment. We tagged additional loci on

the chromosome to evaluate RecN's influence at various distances from the replication fork and thus the DSB. Focus regression occurred at a high frequency near to *oriC* (100 and 450 kb away from *oriC*), less frequently in the middle of the left replichore (1,300 kb) and very infrequently near the terminus (2,300 kb from *oriC*). This suggests that RecN can mediate merging of sister foci upstream from a replication fork but is not able to re-anneal fully segregated chromosomes.

To observe the whole nucleoid dynamics associated with the foci merging, we used cells containing the HU protein fused to mCherry. HU is a histone-like protein that binds ubiquitously to the whole nucleoid. We detected the merging of segregating nucleoids in response to MMC treatment (Fig. 3c,d). This phenomenon was observed 10 ± 10 min after MMC addition and,

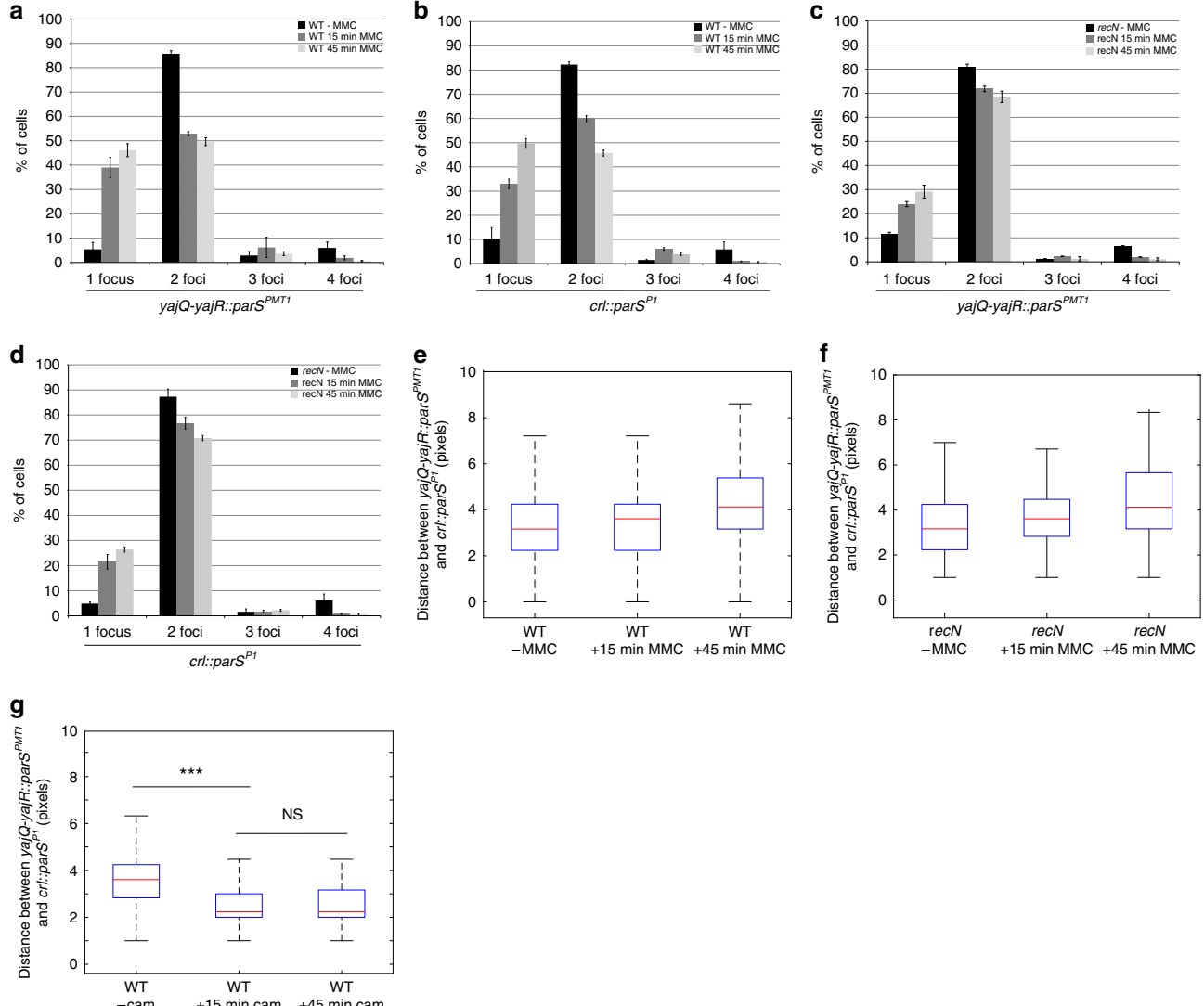

**Figure 4 | RecN influences SCC but not DNA condensation. (a)** The number of foci per cell at the *yajR-yajQ::parS^pMT1* site was counted in cells treated with MMC for 0, 15 or 45 min. **(b)** The number of foci per cell at the *crl::parS^P1* site was counted in cells treated with MMC for 0, 15 or 45 min. **(c)** Same as **a**, but performed in the *recN* mutant. **(d)** Same as **b**, but performed in the *recN* mutant. In **a–d**, error bars represent standard deviations of 300 cells. **(e)** The distance between two loci on the same replichore of the chromosome, spaced by 188 kb and tagged with a *parS^P1* or a *parS^pMT1* site (*crl::parS^P1* and *yajQ-yajR::parS^pMT1*, respectively), was measured after treatment with 10 μg ml⁻¹ MMC for 0, 15 or 45 min. The results are shown as a box plot representing the median, first and forth quartiles (N = 300). **(f)** Same as **e**, but performed in the *recN* mutant. **(g)** The distance between two loci on the same replichore of the chromosome, spaced by 188 kb and tagged with a *parS^P1* or a *parS^pMT1* site (*crl::parS^P1* and *yajQ-yajR::parS^pMT1*, respectively), was measured after treatment with 30 μg ml⁻¹ of chloramphénicol for 0, 15 or 45 min. (*t*-test ***$P < 10^{-30}$, NS (not significant) $P > 10^{-5}$).

as previously described, was dependent on both the RecA and RecN proteins[31]. Regression of sister foci was observed a few minutes following nucleoid merging (Fig. 3e,f and Supplementary Movie 3). These observations suggest that RecN-mediated preservation of SCIs, regression of segregated sister loci and nucleoid merging are coordinated steps in the repair process of DSBs.

**RecN do not promote nucleoid condensation.** The merging of nucleoids observed following MMC treatment may result from two distinct phenomena: a global DNA compaction, mediated by RecN and favoring the random encounter of sister homologues, or an ordered re-zipping that realigns homologous regions of the nucleoid. To distinguish between these two hypotheses, we measured the distance between two loci tagged with *parS^pMT1* and *parS^P1* sites spaced 188 kb apart on the same replichore

(975 kb from *oriC* and 1,163 kb from *oriC*). When cells were treated for 15 min with MMC (at this moment most bi-lobbed cells have merged their nucleoids), the number of foci per cell decreased substantially in the WT strain (1.4 foci per cell on average in the presence of MMC compared to 1.7 in regular conditions, Fig. 4a,b). By contrast, this number remained constant in the absence of RecN (1.65 foci per cell in the presence of MMC compared to 1.7 in regular conditions, Fig. 4c,d). In spite of the reduction of the number of foci, the distance between the tagged loci was unchanged, even though the nucleoids were merged at this time point (Fig. 4e and Supplementary Table 1). After 45 min of MMC application, the distance between the loci increased significantly, reflecting the nucleoid decondensation observed with HU-mCherry and cell filamentation. Importantly, the distance between the two tagged foci was independent on RecN (Fig. 4f). To check that a chromosomal condensation can indeed be observed with our experimental set-up, we performed

the same experiment in the presence of chloramphenicol, an antibiotic that is known to strongly condense the chromosome[40]. The interfocal distance was decreased in the presence of chloramphenicol when compared to untreated cells (Fig. 4g). These observations demonstrate that RecN does not participate in nucleoid condensation and that nucleoid merging is ordered and only leads to encounters between homologous loci. We thus propose that RecN is a cohesion factor that promotes strict realignment of an extensive part of the nucleoid initiating at the site of a damaged replication fork.

**RecN stimulates cell cycle restart after genotoxic stress**. To evaluate the influence of RecN and SCIs on DNA repair efficiency, we analysed cell cycle restart after MMC treatment at the single-cell level on a microfluidic platform. WT and *recN* cells were grown in the microfluidic chamber for 20 min, 10 µg ml$^{-1}$ MMC was injected for 10 min and immediately washed out with fresh medium. In these conditions, the WT and *recN* strains show almost similar viability to the untreated cells. We can thus observe the impact of RecN on the efficiency of repair rather than viability. Cell division and nucleoid dynamics were followed for 3 hours after washing (Fig. 5a,b and Supplementary Movies 4 and 5). In the presence of RecN, 70% of the bacteria recovered from the MMC treatment and underwent one (9%), two (50%) or three divisions (41%) in the ensuing 2 h. In the absence of RecN, only 36% of the cells recovered and performed one (16%), two (59%) or three divisions (25%). The number of filamenting cells at the end of the time course was also strongly reduced in the presence of RecN (15% in WT cells compared to 56% in the *recN* mutant). Altogether, these results suggest that RecN activity contributes to accelerate the repair process and thus allows a rapid return to normal growth.

**The absence of RecN modifies RecA dynamics**. It has been proposed that RecA contributes to RecN loading onto nucleoids via a direct RecA–RecN interaction[23], and our results suggest that RecN favors the rapid repair of MMC-induced lesions. We therefore sought to determine whether RecN influences RecA-mediated homology search and RecA-mediated DNA repair by preserving SCIs. RecA forms repair foci in the cell in the presence of DNA damage[41], and the presence of Rad51 or 52 repair foci in eukaryotic cells is considered a good reporter of ongoing DNA repair[42]. Thus, we performed time-lapse fluorescence microscopy in the presence of MMC in strains containing an ectopic *recA*-mCherry fusion in addition to the wild-type *recA* gene. RecA-mCherry formed large aggregated foci at the pole as well as small dynamic foci that likely correspond to repair foci (Fig. 5c). The repair foci were very dynamic, and their fluorescence was weak. In the continuous presence of MMC, these foci only persisted at a given position for 10–20 min (Fig. 5e). In the absence of RecN, RecA formed foci and elongated dynamic structures (Fig. 5d–f). Elongated structures were observed in 21% of *recN* cells at any given time point (Fig. 5g), but almost every cell presented one at some point during the 90 min time course. They persisted for 30 ± 10 min. Such structures, called RecA bundles, have been described following sister chromatid cleavage by I-SceI, although their role in recombination repair is not yet understood[18]. However, in contrast with this previous report, we observed very few bundles in WT cells after MMC treatment, suggesting that bundles form preferentially when the broken sister is far from its intact homologue.

## Discussion

Repair of DNA damage by homologous recombination requires the presence of an undamaged sister homologue. In *E. coli*, during a regular cell cycle, sister chromatids are kept in close contact by topological links called precatenanes[16,17,43]. They allow for perfect alignment of sister chromatids and thus promote site-specific recombination between sister loci[16]. SCIs are thought to favour homologous recombination and could thus accelerate the repair process. However, because topological structures diffuse extensively on DNA, topological cuffing of sister chromatids might not persist if DNA is broken. In the present work, we unravel a role for SCIs in DNA damage repair induced by MMC. Furthermore, their preservation in the presence of MMC requires induction of the SOS response. We demonstrate here that RecN is a central protein for the preservation of SCIs (Fig. 1). The lack of *recN* is not as detrimental as *recA* or *lexA ind-* mutants for SCIs preservation, suggesting that another yet unknown factor may also participate in the process.

RecN is a well-conserved bacterial SMC protein, and its involvement in the repair of DSBs has been known for some time. In *E. coli,* RecN expression is repressed under regular growth conditions but is strongly expressed by the SOS regulon in the presence of MMC, ultraviolet, quinolone drugs or oxidative stress[7,44]. Based on research in *D. radiodurans* and *B. subtilis*, two different functions have been proposed for RecN: a cohesion function[27] and an end-joining function[26,28]. Recent work has suggested that RecN loading onto DSBs requires interaction with RecA[23]. In our study, we demonstrate that RecN induction allows for preservation of SCIs and abolishes segregation of newly replicated loci (Fig. 1). In theory, preservation of SCIs may be possible if the binding of RecN to DNA ends prevents precatenane diffusion through the DSB. Importantly, because the absence of RecN can be rescued by a mutation that affects Topoisomerase IV function, we propose that RecN bridges sister chromatids in a manner similar to cohesins (Fig. 6). We cannot exclude that RecN participates in DSB end joining in *E. coli,* however our observations demonstrate that end joining is not the only function of RecN in *E. coli*. This is in good agreement with the huge amount of RecN produced upon SOS induction[7,22]. ChIP-seq experiments demonstrated that RecA filaments extend over 20 kb from a double-strand break, this is much less than the extent of the genome affected by RecN activity (SCI preservation and the regression of segregated sister foci) suggesting that even if RecA directly interacts with RecN near the DSB, RecN should be able to escape and propagate on the genome (Fig. 6). Such a process is reminiscent with what has been observed for other bacterial SMC proteins, such as MukB and SMC (*B. subtilis* or *C. crescentus*) that are respectively loaded by MatP[45] and ParB[46–49]. The mechanism by which RecN promotes cohesion is not yet understood. As *E. coli* RecN is a small SMC protein (∼1/3 of SMC3), it is unlikely that a dimer alone could form a ring that entraps two DNA double strands. Multimers of of *D. radiodurans* RecN have been observed, therefore, we can postulate that head-to-tail RecN multimerization can favour long stretches of sister chromatid pairing (Fig. 6).

Theoretically, replication-dependent precatenanes could facilitate repair via homologous recombination by increasing SCIs. However, the observations that RecN is required to preserve SCIs in the presence of MMC suggest that precatenanes are not sufficient to maintain SCIs under these conditions. We observed that Topo IV alteration, which prevents the removal of post-replicative precatenanes, can fully compensate for the absence of RecN in the presence of MMC. This means that when post-replicative precatenanes are not efficiently removed by Topo IV, they facilitate homology search and homologous recombination. It has been proposed that precatenanes do not accumulate homogeneously along the chromosome[15,16], some regions called SNAPs tend to remain colocalized, presumably because SeqA bound to these regions inhibits Topo IV activity[43]. SCIs are

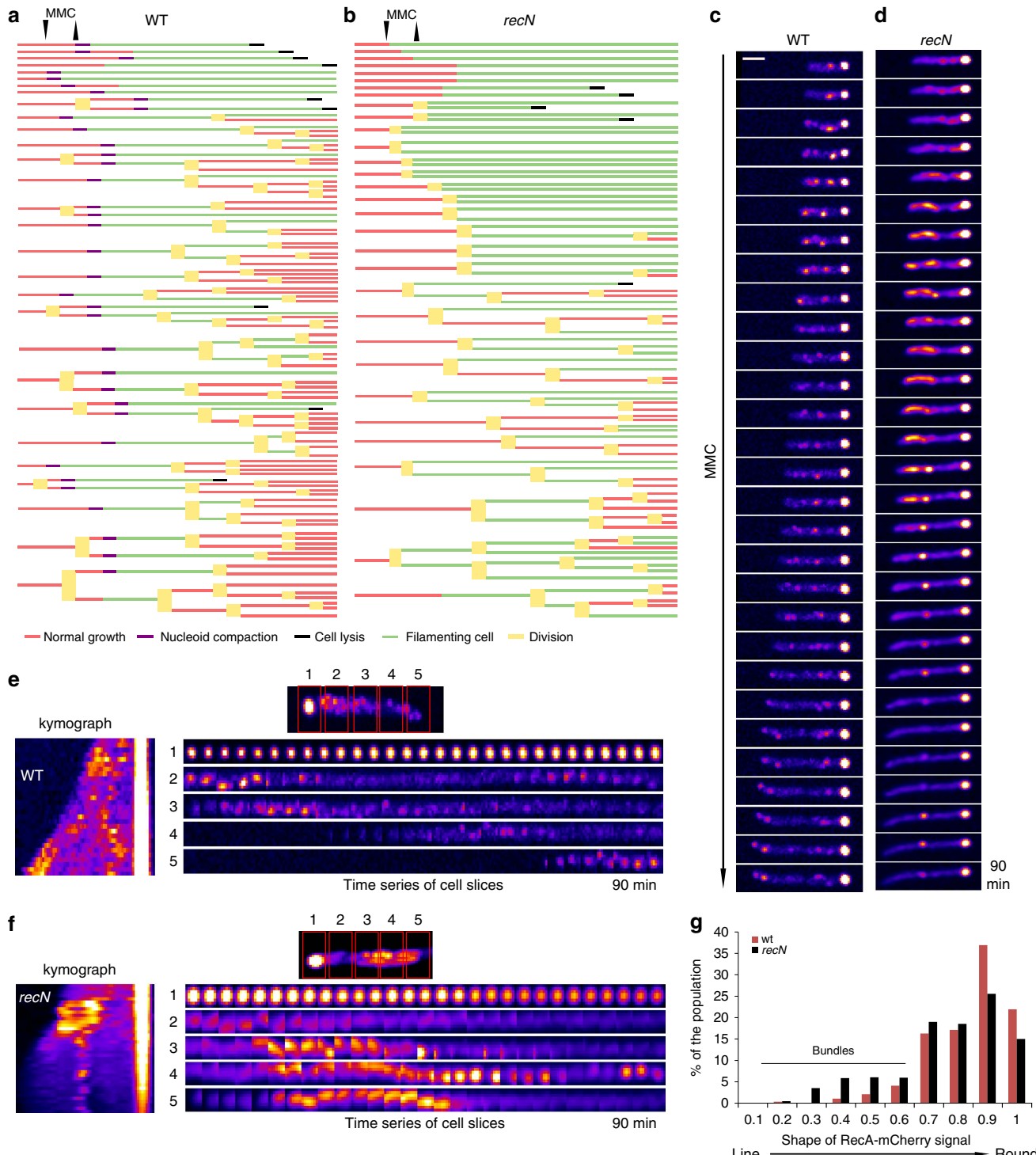

Figure 5 | *recN* mutant has altered homology search and delayed cell cycle restart. (**a**) Cell cycle restart pattern after a brief MMC treatment using a microfluidic platform. WT cells were introduced into the chambers, and fresh minimal medium A was perfused for 20 min; 10 µg ml$^{-1}$ MMC was then perfused for 10 min and immediately washed with clean medium, and incubation and imaging were continued for over 3.5 h. Cell lineage was measured for 100 cells; each colour corresponds to a given state of the cell. (**b**) The same experiment as described in **a** was performed in the *recN* mutant, which exhibits delayed cell cycle restart. (**c**) Representative time-lapse microscopy of RecA-mCherry focus dynamics in the presence of MMC in the WT strain. Time-lapse imaging starts at 5 min after initial contact with MMC. Pictures were acquired every 3 min for 2 h on an agarose pad with MMC. (**d**) RecA-mCherry focus dynamics in the presence of MMC in the *recN* mutant. Experiments were performed as described for **c**. (**e**) Analysis of RecA focus dynamics. Kymograph and time series of cell slices for RecA-mCherry WT. The experiment was performed as described in **c**. (**f**) Analysis of RecA foci dynamics in the *recN* mutant. The experiment was performed as described in **d**. (**g**) The frequency of bundles was estimated as a function of the shape of the RecA mCherry signal in WT and the *recN* mutant. The WT and *recN* distribution are significantly different (*t*-test $P = 10^{-30}$). Scale bar is 1 µm.

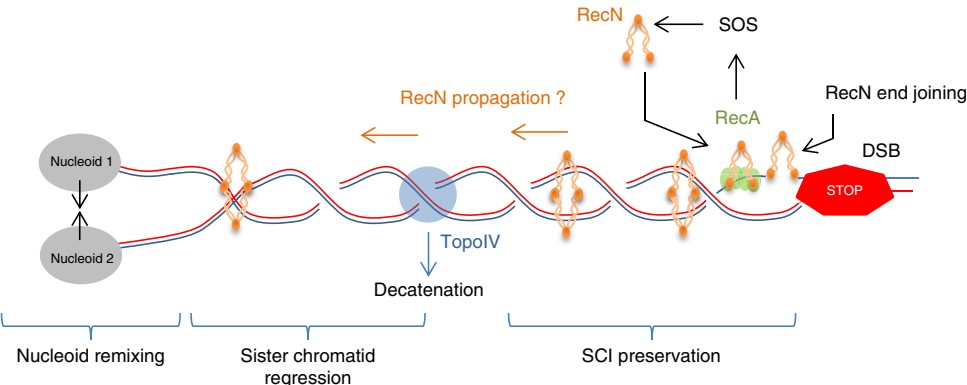

**Figure 6 | Roles of RecN during repair of an induced DSB.** Our observations suggest that when a replicative DSB occurs, RecA (Green) is responsible for RecN (orange) expression (through the SOS response) and RecN loading onto the sister chromatids. RecN loading prevents the complete removal of SCIs by Topo IV (blue) and may participate in DNA end joining. In a second step, RecN may propagate on the newly replicated chromatids to mediate regression of the segregated sister chromatids and re-mixing of brother nucleoids.

preserved in SNAPs, and it would be interesting to ascertain whether this corresponds to improved homologous recombination at these sites.

In addition to preserving SCIs, MMC provokes regression of segregated sister foci. Similar observations have been previously reported for I-SceI[18,19,31]. This regression of sister chromatids over a large distance is correlated with the re-merging of segregating nucleoids. SCI preservation, sister regression and nucleoid merging are dependent on the RecA and RecN proteins. The fluorescent labelling of two loci spaced 188 kb apart on the same replichore revealed that MMC induces RecN-dependent realignment of the nucleoids rather than random condensation. Our observations suggest that inhibition of sister segregation is the first activity of RecN when recruited to the DSB by RecA and that segregated loci regression is a secondary step. However, we do not yet understand the mechanism promoting this or the purpose of such a profound chromosomal reorganization.

Bacteria experience a large number of stresses in the environment or in their host, many of which induce DSBs. To survive DSBs, *E. coli* induces the SOS response, which blocks cell division and suspends the bacterial cell cycle. Following genotoxic stress, it is essential for damaged bacteria to restart growth as quickly as possible. Therefore, SOS induction could be essential for survival in an environment in which competition among bacterial species is high. Our observations demonstrate that growth recovery is significantly accelerated in the presence of RecN (Fig. 5a,b), and this capacity of RecN to accelerate repair might be responsible for its high conservation among bacteria. In this way, RecN can be viewed as an ancestor of the SMC5/6 complex that maintains stalled replication forks in a recombination-competent conformation[50,51].

## Methods

**Strains.** The strains and plasmids used in this study are described in Supplementary Table 1. All strains are derived from wild-type MG1655 or MG1656 (Δlac MluI). Strains containing *loxP* sites were constructed by λ red recombination using the plasmid pGBKDlaclox as matrix[16]. The strains used for microscopy were constructed by λ red recombination using the plasmid pGBKDparS-pMT1 as matrix[52]. Details of the construction are presented in the Supplementary Methods. The RecA-mCherry fusion is a gift from Bénédicte Michel; its construction is described in the Supplementary Methods.

***LoxP* assays.** Every experiment is performed in the same conditions. An overnight culture was diluted 1:200 in Minimum Media A supplemented with 0.2% glycerol and 0.2% casamino acids. Three to five biological replicates where performed for each sample. The cells were grown at the indicated temperature to an $OD_{600nm}$ of ∼0.2. In these conditions, generation time at 37 °C is 65 min. Cre expression was induced by the addition arabinose (0.1%) to growth media. Genotoxic stress was

induced by the addition of $10 \mu g \, ml^{-1}$ MMC to the growth media at time point 0. At each time point, 1.5 ml of cells was flash frozen in liquid nitrogen. Genomic DNA was extracted using the Pure Link Genomic DNA Mini Kit (Life Technologies) and quantified using a Nanodrop spectrophotometer (Thermo Scientific). Genomic DNA was diluted to $2 \, ng \, ml^{-1}$, and PCR was performed using ExTaq polymerase (Takara). The amplified DNA was analysed using the DNA 1000 Assay on a Bioanalyzer (Agilent). The frequency of recombination was measured as follows: (amount of 1*loxP* DNA + amount of 2*loxP* DNA)/(total amount of *loxP* DNA).

**Colony forming unit measurement.** At an $OD_{600nm}$ of 0.2, $10 \mu g \, ml^{-1}$ MMC or $1 \mu g \, ml^{-1}$ AZT was added to the culture. Cell viability was followed every 10 min for 40 min. At each time point, cells were serially diluted in LB ($10^0$–$10^{-6}$) and plated on LB agar plates. Plates were incubated for 16 h at 37 °C, and colonies were counted.

**EdU staining.** At an $OD_{600nm}$ of 0.2, the cells were either incubated with $10 \mu g \, ml^{-1}$ of MMC for 10 min and then incubated with an equal amount of 2X EdU (Click-IT Assay Kit, Thermo Fisher Scientific) (to monitor DNA replication) or first incubated with an equal amount of 2X EdU for 10 min and then MMC (to monitor DNA degradation). Cells were then fixed with Formaldehyde mix (5% Formaldehyde, 0.05% Glutaraldehyde and 1 × PBS) for 10 min at room temperature and 50 min on ice. Cells were washed three times with 1 × PBS and resuspended in 98 μl of fresh GTE buffer (50 mM glucose, 20 mM Tris pH 8, 10 mM EDTA). Cells can be left ON at 4 °C. Following this step, 2 μl of freshly prepared $500 \mu g \, ml^{-1}$ lysozyme were added to each sample and the sample was then immediately transferred onto a poly-lysine-coated slide and left for 3 min at room temperature. The slide was rinsed twice with 1 × PBS, and 150 μl Click-It Cocktail (prepared according to manufacturer's instructions) was added; the sample was then incubated in the dark at room temperature for 30 min. The slides were rinsed two times with 1 × PBS + DAPI ($1 \mu g \, ml^{-1}$). The slides were then rinsed 10 times with 1 × PBS and left to dry at room temperature. Finally, 10 μl of SlowFade (Thermo Fisher Scientific) were added to the slide. The slides were stored at 4 °C for one hour before imaging.

**Microscopy.** An overnight culture was diluted 1:200 in Minimum Media A supplemented with 0.2% casamino acids and 0.25% glucose (to limit overexpression of ParB protein glucose is used instead of glycerol for microscopy experiments). In these conditions, generation time at 37 °C is 50 min. The cells were grown to an $OD_{600nm} = 0.2$ at 37 °C, pelleted and resuspended in 50 μl of fresh medium. One per cent Agarose pad slides were prepared within a gene frame (VWR)[53]. Genotoxic stress was induced by the addition of $10 \mu g \, ml^{-1}$ MMC or $1 \mu g \, ml^{-1}$ AZT to the agarose pad. Time-lapse microscopy was performed using a confocal spinning disk (X1 Yokogawa) on a Nikon Ti microscope at 100 × magnification controlled by Metamorph (Molecular Imaging) and an EMCCD camera (Roper). Definite focus (Nikon) was used for each time point. Images were acquired every 3 min for 2 h at 30 °C. Five positions were observed simultaneously for each experiment, with 20–50 cells per position. Snapshot experiments for focus counting and inter-foci distance measurements were performed as previously described[53].

**Microfluidic experiments.** An overnight culture was diluted 1:200 in Minimum Media A supplemented with 0.2% casamino acids and 0.25% glucose. The cells were grown to an $OD_{600nm}$ of approximately 0.2. A microfluidic plate was set-up according to the Merck Millipore protocol for bacteria. Medium changes were

controlled by the Onyx system from Merck Millipore. Fresh minimum medium A was perfused for 20 min at 37 °C; 10 μg ml$^{-1}$ MMC was perfused for 10 min and fresh medium was perfused for 3 h. Images were acquired every 3 min using a confocal spinning disk (Yokogawa W1) on a Zeiss Axio imager microscope at a × 63 magnification with an Orca Flash 4 camera (Hamamatsu). Time-lapse images were acquired using Metamorph (Molecular Imaging) and analysed with ImageJ software.

**Data availability.** All relevant data, material and methods are available from the authors.

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

## Acknowledgements

We thank Bénédicte Michel and Michele Valens for the gift of the strains. We also thank Bénédicte Michel for scientific advice and careful reading of the manuscript. We thank Anouar Khayachi for his help on the Co-Immunoprecipitation experiments. We thank Avradip Chatterjee and Kenneth Marians for RecN antibody. We thank Angela Taddei for helping us with the Onix Cell microfluidics system. This work was supported by ANR (grants ANR-14-CE10-0007 MAGISBAC and ANR-15-CE11-0023-01 HIRESBACS), Fondation ARC, Ligue contre le Cancer and Fondation pour la Recherche Medicale.

## Author contributions

E.V., C.P., I.G.J., C.C. designed, performed and analysed experiments; O.E. designed and analysed experiments. E.V., C.C. and O.E. wrote the manuscript.

## Additional information

**Competing financial interests:** The authors declare no competing financial interests.

