## [Peer Review File · Nature Communications]

Reviewers' Comments:

Reviewer #1 (Remarks to the Author)

Comments to the Authors:

In the paper by Vickridge et al submitted to Nature Communications, the authors investigate the possibility that the E coli SMC protein RecN is important for sister chromatid cohesion induced by genotoxic stress, i e for Damage induced cohesion. Damage induced cohesion has so far mainly been studied in *S cerevisiae* by a limited number of labs. It is therefore tremendously interesting and might add important knowledge and understanding on the concept, as well as new possibilities to understand why and how this phenomenon is activated, created and used by cells for DNA repair and/or general genome maintenance. It is also rewarding that this mechanism seems to be so well conserved.

I have read the paper with interest and have no major issues or concerns regarding the execution of the experiments or the conclusions drawn from them, they are both convincing and well controlled. I have only a number of minor comments.

First, in the introduction when talking about cohesion in relation to DSB repair, please check the referencing. Sjögren and Nasmyth 2001 showed that the cohesion established during S phase is important for DSB repair in G2, which is not clear from the phrasing. Loading of Cohesin to DSBs in yeast was shown both by both Unal et al and Strom et al 2004 in Mol Cell (back to back). In the same paper Strom et al showed that cohesion (new G2 specific sister chromatid interaction) was actually established post replication in response to DNA damage. Loading of cohesin genome wide post replication was shown very early by the Nasmyth and Koshland groups, but at that time it was concluded that these cohesin molecules could not establish cohesion (as mentioned later found to be the case in response to damage only). Formation of cohesion genome wide, on undamaged chromosomes in response to a DSB, was shown by Unal et al and Strom et al in Science 2007 (back to back). Please change accordingly, depending on what you want to emphasize.

Second, some typos: should it be SCC or SCI in Figure Legend 1, line 719? a legend for the lines in Fig 3F would be helpful.

And finally, the CoIP experiment for RecA and RecN in Supplementary figure 3, is not at all very convincing. If this is a critical statement I think this experiment should be worked on to become clear. What dark spots are bands and what is just background or unspecific binding to Abs or beads?

Reviewer #2 (Remarks to the Author)

The manuscript by Vickridge et al addresses the role of sister chromosome proximity in the repair of damaged DNA in *E. coli*. Using a previously established DNA recombination assay, the authors show that sister chromosome interactions (SCI) are maintained for significant periods of time after exposure of cells to DNA damaging agents, while SCIs are rapidly lost after inhibition of DNA replication. Maintenance of SCI depends on RecA and the induction of the SOS response but apparently not on successful repair of DNA damage by RecA driven homologous recombination. Normal levels of SCI maintenance moreover require an intact *recN* gene, which is highly upregulated during the SOS response. Intriguingly, SCI loss in a *recN* mutant can be rescued by inactivation of topoisomerase IV. The latter suggests that RecN helps to maintain SCI by

preventing decatenation of sister chromosomes by topoisomerase IV. The authors provide evidence that RecN might moreover be involved in the re-joining of sister chromosomes behind the replication fork (upon DNA damage), possibly thereby promoting the search for homologous DNA repair templates.

The manuscript covers an impressive body of work and clearly advances the field by defining and characterizing a previously postulated function for RecN and for maintenance of SCI for homologous recombination. The conclusions of the paper are valid, although in some instances slightly overstated (see below). With some modifications, the paper will be highly valuable for a larger research community and should be a great candidate for publication in Nature Communications.

Major points

The finding that topoIV inhibition alleviates the lethality of MMC treatment for recN mutants is striking and surprising. However, from the presented data it remains unclear how specific the suppression is for RecN function. It would be highly desirable to include other rec mutants and test for suppression of lethality by topoIV inactivation. Either result would strengthen the manuscript significantly and will help to avoid illegitimate conclusions.

The cross-link co-IP experiments shown in Figure S3C and S3D do need negative control samples (for example un-tagged RecN for S3C). In the current form, little can be concluded from these experiments. Since the data is not directly relevant for the main conclusions of the paper (and since an interaction has been suggested previously), it might be omitted altogether.

Abstract, line 10-13. This sentence is unnecessarily over-stating the findings of the manuscript. 'fully compensated' should be softened. The double mutant displays high lethality compared to wild type. Likewise, 'demonstrating' in the same sentence should be replaced by for example 'being consistent'. Page 14, line 398: The data in the manuscript goes short of 'showing' that the preservation of SCIs is an 'essential' process. It merely reveals a correlation between loss of SCIs and lethality upon MMC treatment. The conclusion needs to be softened. Similarly, on page 10, line 252, 'demonstrates' should to be reworded. Page 9, line 255: The manuscript does not address the mechanistic basis of RecN function. Thus, it remains unclear, whether RecN plays a 'structural' role. Please reword the headline.

Minor comments

Some references are missing from the reference list. For example references in: page 15, line 437 and 438.

Figure 1A: what does the abbreviation 'SC' stand for?

In figure 2B and 2D, the axis labelling is incomplete. Is the CFU given relative to time point zero in log₁₀ scale (as in figure S3A)?

In figure 2E, labels or legends for the gray and black bars are missing.

Figure 4G might better be moved to supplemental figures or included in Figure 4C and 4F. In its current display, the data is quite difficult to digest.

Label for panel 'D' in figure S3 is mis-placed.

Page 5, line 101. 'promising' sounds strange here. 'intriguing' or 'interesting' might be better fitting.

Page 3, line 21. 'maintain their DNA' could be replaced by 'maintain the integrity of their DNA'?

Management of *E. coli* sister chromatid cohesion in response to genotoxic stress

Elise Vickridge^{1,2,3}, Charlene Planchenault¹, Charlotte Cockram¹, Isabel Garcia Junceda¹, and Olivier Espéli^{1*}

Response to reviewers' comments:

Reviewer #1 (Remarks to the Author):

Comments to the Authors:

In the paper by Vickridge et al submitted to Nature Communications, the authors investigate the possibility that the *E. coli* SMC protein RecN is important for sister chromatid cohesion induced by genotoxic stress, i.e. for Damage induced cohesion. Damage induced cohesion has so far mainly been studied in *S. cerevisiae* by a limited number of labs. It is therefore tremendously interesting and might add important knowledge and understanding on the concept, as well as new possibilities to understand why and how this phenomenon is activated, created and used by cells for DNA repair and/or general genome maintenance. It is also rewarding that this mechanism seems to be so well conserved.

We appreciate the interest that the reviewer shows for our work.

I have read the paper with interest and have no major issues or concerns regarding the execution of the experiments or the conclusions drawn from them, they are both convincing and well controlled. I have only a number of minor comments.

First, in the introduction when talking about cohesion in relation to DSB repair, please check the referencing. Sjögren and Nasmyth 2001 showed that the cohesion established during S phase is important for DSB repair in G2, which is not clear from the phrasing. Loading of Cohesin to DSBs in yeast was shown both by both Unal et al and Strom et al 2004 in Mol Cell (back to back). In the same paper Strom et al showed that cohesion (new G2 specific sister chromatid interaction) was actually established post replication in response to DNA damage. Loading of cohesin genome wide post replication was shown very early by the Nasmyth and Koshland groups, but at that time it was concluded that these cohesin molecules could not establish cohesion (as mentioned later found to be the case in response to damage only). Formation of cohesion genome wide, on undamaged chromosomes in response to a DSB, was shown by Unal et al and Strom et al in Science 2007 (back to back). Please change accordingly, depending on what you want to emphasize.

We agree with the reviewer this paragraph was un clear, We have thoroughly modified it.

In eukaryotes, during replication, large multi-protein complexes called cohesins keep the newly replicated sister chromatids together before segregation (Nasmyth and Haering, 2009). Cohesin-mediated pairing of sister loci is important for DSB repair in G2 phase (Sjögren and Nasmyth,

2001), perhaps for initiating the repair process and favoring homologous recombination. Post-replicative recruitment of cohesins has been observed at the site of the DSB. In unperturbed cells, cohesin loaded following at the postreplicative phase is not capable of generating cohesion. However, cohesin loaded at a DSB generates de novo G2 cohesion (Ström et al., 2004). The DSB-induced cohesion is not limited to broken chromosomes but occurs also on unbroken chromosomes, suggesting that cohesion provides genome-wide protection of chromosome integrity (Ström et al., 2007; Unal et al., 2007).

Second, some typos: should it be SCC or SCI in Figure Legend 1, line 719? a legend for the lines in Fig 3F would be helpful.

These points have been modified as suggested.

And finally, the CoIP experiment for RecA and RecN in Supplementary figure 3, is not at all very convincing. If this is a critical statement I think this experiment should be worked on to become clear. What dark spots are bands and what is just background or unspecific binding to Abs or beads?

The Co-IP experiment (figure S5) has been worked on. Co-immunoprecipitation of RecN with a RecA antibody is robust and specific to MMC treated cells. Co-immunoprecipitation of RecA with RecN-Flag with a Flag antibody is less specific, presumably because of RecA-beads non-specific interactions. Nevertheless in the presence of MMC and RecN-Flag induction the amount of co-immunoprecipitated RecA significantly increased.

Reviewer #2 (Remarks to the Author):

The manuscript by Vickridge et al addresses the role of sister chromosome proximity in the repair of damaged DNA in E. coli. Using a previously established DNA recombination assay, the authors show that sister chromosome interactions (SCI) are maintained for significant periods of time after exposure of cells to DNA damaging agents, while SCIs are rapidly lost after inhibition of DNA replication. Maintenance of SCI depends on RecA and the induction of the SOS response but apparently not on successful repair of DNA damage by RecA driven homologous recombination. Normal levels of SCI maintenance moreover require an intact recN gene, which is highly upregulated during the SOS response. Intriguingly, SCI loss in a recN mutant can be rescued by inactivation of topoisomerase IV. The latter suggests that RecN helps to maintain SCI by preventing decatenation of sister chromosomes by topoisomerase IV. The authors provide evidence that RecN might moreover be involved in the re-joining of sister chromosomes behind the replication fork (upon DNA damage), possibly thereby promoting the search for homologous DNA repair templates.

The manuscript covers an impressive body of work and clearly advances the field by defining and characterizing a previously postulated function for RecN and for maintenance of SCI for homologous recombination. The conclusions of the paper are valid, although in some instances slightly overstated (see below). With some modifications, the paper will be highly valuable for a larger research community and should be a great candidate for publication in Nature Communications.

We appreciate the interest that the reviewer shows for our work.

Major points

The finding that topoIV inhibition alleviates the lethality of MMC treatment for recN mutants is striking and surprising. However, from the presented data it remains unclear how specific the suppression is for RecN function. It would be highly desirable to include other rec mutants and test for suppression of lethality by topoIV inactivation. Either result would strengthen the manuscript significantly and will help to avoid illegitimate conclusions.

The reviewer raised an interesting point. To test if TopoIV alteration specifically rescues the viability of recN mutant we performed the same assay for the recA and lexAind- mutants. The results are presented on Fig 2B, Supplementary Fig 3G and 3H. We performed the CFU in the recA mutant at two different MMC concentrations (10 and 2 µg/ml) to eventually observe a TopoIV alteration influence in the presence of a small amount of damages. Topo IV alteration did not rescue recA deletion. Similar results were obtained for the lack of SOS induction in the lexAind- strain.

The cross-link co-IP experiments shown in Figure S3C and S3D do need negative control samples (for example un-tagged RecN for S3C). In the current form, little can be concluded from these experiments. Since the data is not directly relevant for the main conclusions of the paper (and since an interaction has been suggested previously), it might be omitted altogether.

The Co-IP experiment (figure S5) has been worked on. Co-immunoprecipitation of RecN with a RecA antibody is robust and specific to MMC treated cells. Co-immunoprecipitation of RecA with RecN-Flag with a Flag antibody is less specific, presumably because of RecA-beads non-specific interactions. Nevertheless in the presence of MMC and RecN-Flag induction the amount of co-immunoprecipitated RecA significantly increased.

Abstract, line 10-13. This sentence is unnecessarily over-stating the findings of the manuscript. 'fully compensated' should be softened. The double mutant displays high lethality compared to wild type. Likewise, 'demonstrating' in the same sentence should be replaced by for example 'being consistent'. Page 14, line 398: The data in the manuscript goes short of 'showing' that the preservation of SCIs is an 'essential' process. It merely reveals a correlation between loss of SCIs and lethality upon MMC treatment. The conclusion needs to be softened. Similarly, on page 10, line 252, 'demonstrates' should to be reworded.

These sentences have been modified as suggested by the reviewer. In the abstract: "The loss of sister chromatid interactions and viability defects observed in the absence of RecN were compensated by alterations in topoisomerase IV, suggesting that the main role of RecN during DNA repair is to promote contacts between sister chromatids". In the result section (line 259): "This suggests that, in this context, maintaining precatenanes behind the replication fork can compensate for the absence of RecN, and that more generally, in spite of a DSB most precatenane links do not immediately disappear." and (line 283): "These observations strengthen the hypothesis that topological links, when they are artificially maintained, compensate for a lack of RecN and therefore suggests that RecN is playing a structural role by maintaining sister chromatids close together."

Page 9, line 255: The manuscript does not address the mechanistic basis of RecN function. Thus, it remains unclear, whether RecN plays a 'structural' role. Please reword the headline.

The title of the paragraph has been changed to : " The lack of RecN can be compensated by excess sister chromatid precatenation"

Minor comments

Some references are missing from the reference list. For example references in: page 15, line 437 and 438.

Figure 1A: what does the abbreviation 'SC' stand for?

In figure 2B and 2D, the axis labelling is incomplete. Is the CFU given relative to time point zero in log₁₀ scale (as in figure S3A)?

In figure 2E, labels or legends for the gray and black bars are missing.

Figure 4G might better be moved to supplemental figures or included in Figure 4C and 4F. In its current display, the data is quite difficult to digest.

Label for panel 'D' in figure S3 is mis-placed.

Page 5, line 101. 'promising' sounds strange here. 'intriguing' or 'interesting' might be better fitting.

Page 3, line 21. 'maintain their DNA' could be replaced by 'maintain the integrity of their DNA'?

All these mistakes or imprecisions have been corrected.

Reviewers' Comments:

Reviewer #1 (Remarks to the Author)

In this revised version of the manuscript; Management of E. coli sister chromatid cohesion in response to genotoxic stress, submitted by the Olivier Espeli group, the authors have responded satisfyingly to my questions and concerns, and I have no further issues to address.

Reviewer #2 (Remarks to the Author)

The authors have carefully revised the manuscript according to the reviewer's comments. The manuscript should now be ready for publication.

One minor comment: The authors include two additional mutations to characterize the compensation by topo IV inhibition. The mutants do not show suppression of MMC sensitivity, however, both of them have a very strong phenotype (compared to recN) and might thus not be the ideal choice for the experiment. Possibly, in the future other mutants should be investigated as well.